# Non Monotonous Effects of Noncovalently Functionalized Graphene Addition on the Structure and Sound Absorption Properties of Polyvinylpyrrolidone (1300 kDa) Electrospun Mats

**DOI:** 10.3390/ma12010108

**Published:** 2018-12-30

**Authors:** Giuseppe Rosario Del Sorbo, Greta Truda, Aurelio Bifulco, Jessica Passaro, Giuseppe Petrone, Bonaventura Vitolo, Giovanni Ausanio, Alessandro Vergara, Francesco Marulo, Francesco Branda

**Affiliations:** 1Department of Chemical Materials and Industrial Production Engineering (DICMaPI), University of Naples Federico II, P.le Tecchio 80, 80125 Naples, Italy; giuppyro@virgilio.it (G.R.D.S.); aurelio.bifulco@unina.it (A.B.); j.passaro89@gmail.com (J.P.); 2Department of Industrial Engineering, Aerospace Division, University of Naples Federico II, Via Claudio 21, 80125 Naples, Italy; gretatruda@hotmail.it (G.T.); giuseppe.petrone@unina.it (G.P.); francesco.marulo@unina.it (F.M.); 3GEVEN S.p.A., Via Boscofangone, Naples, loc. ASI, 80035 Nola, Italy; bonaventura.vitolo@geven.com; 4Department of Physical Sciences, University of Naples Federico II, Via Cinthia 21, 80126 Naples, Italy; giovanni.ausanio@unina.it; 5Department of Chemical Sciences, University of Naples Federico II, Via Cinthia 21, 80126 Naples, Italy; alessandro.vergara@unina.it

**Keywords:** electrospinning, polyvinylpyrrolidone, graphene, Raman, acoustical proprieties

## Abstract

Graphene is an attractive component for high-performance stimuli-responsive or ‘smart’ materials, shape memory materials, photomechanical actuators, piezoelectric materials and flexible strain sensors. Nanocomposite fibres were produced by electrospinning high molecular weight Polyvinylpyrrolidone (PVP-1300 kDa) in the presence of noncovalently functionalised graphene obtained through tip sonication of graphite alcoholic suspensions in the presence of PVP (10 kDa). Bending instability of electrospun jet appears to progressively increase at low graphene concentrations with the result of greater fibre stretching that leads to lower fibre diameter and possibly conformational changes of PVP. Further increase of graphene content seams having the opposite effect leading to greater fibre diameter and Raman spectra similar to the pure PVP electrospun mats. All this has been interpreted on the basis of currently accepted model for bending instability of electrospun jets. The graphene addition does not lower the very high sound absorption coefficient, α, close to unity, of the electrospun PVP mats in the frequency range 200–800 Hz. The graphene addition affects, in a non-monotonous manner, the bell shaped curves of α versus frequency curves becoming sharper and moving to higher frequency at the lower graphene addition. The opposite is observed when the graphene content is further increased.

## 1. Introduction

Recently [1,2] the production of polymer/nanocarbons composites through electrospinning raised enormous interest. Generally speaking, it is well known [1] that the research activity on nanocomposites grew very much particularly in the fields of ultralight and ultrastrong composites, renewable energy harvesting and storage and bio-medical materials [1,2,3,4,5]. It was also pointed out that nanocarbon reinforced composites may combine the positive attributes of the host and reinforcement materials in novel ways [6,7,8].

Graphene is one of the most promising materials because of its large specific surface area, good electrical conductivity, high Young’s modulus and thermal conductivity [9,10,11]. Mechanical strength, thermal and electrical conductivity may be largely increased with the addition of graphene to both polymeric than inorganic systems [1]. Graphene is an attractive component for high-performance stimuli-responsive or ‘smart’ materials [12,13]. Graphene is used [14] in shape memory materials, photomechanical actuators and piezoelectric [15] materials. Graphene/polyurethane based flexible strain sensors were recently proposed [16].

Electrospinning allows to prepare very thin polymeric nanofibers down to a few nanometres, [3,4,5,17,18,19,20,21,22,23,24,25]. Many electrospun functional polymer nanofibers were reinforced with nanocarbons, [1,2]. Carbon nanotubes and graphene, can, also in the case of electrospun mats, significantly enhance the mechanical, electrical and thermal properties, leading to promising applications in biology and sensors [7]. Electrospun ultrafine graphene fibres were proposed for use in electrodes, conductive wires, smart fabrics and similar applications, which require conductive filler inside the polymer nanofibers [13].

Recently the authors demonstrated [26] that electrospinning may be useful to produce very interesting soundproofing materials. Noise pollution is one of the most widespread irritations and constitutes real danger to the human health [27,28,29,30] with numerous, pervasive, persistent and medically and socially significant effects that impair health and degrade residential, social, working and learning environments with corresponding real (economic) and intangible (wellbeing) losses [31]. The authors demonstrated [26] that electrospinning allows to produce sound absorbers with reduced thickness (2–3 cm) and excellent sound-absorption properties in the low and medium frequency range [26] where the traditional porous materials fail, human sensitivity is high and interest in the transportation industry is great [29,30,31,32].

In the present work, light non-woven mats (density = 63 kg/m^3^) were successfully obtained by electrospinning PVP of high molecular weight (1300 KDa), at different graphene concentration. The interest in the addition of graphene comes from the possibility it offers, in combination with other inorganic nanomaterials, to improve flame retardancy [33] that often, because of very severe regulations (i.e., in aerospace engineering), prevents their applicability [34]. For some applications also the expected improvements of mechanical strength could be of interest.

The blankets were prepared by electrospinning solutions of PVP of high molecular weight (1300 KDa) in which noncovalently functionalized graphene had been dispersed. It is well known [1,13], in fact, that dispersing and aligning the nanofillers in a polymer matrix is very difficult owing to the strong Van der Waals attractions between nanocarbons that induce aggregation and prevent dispersion. The problem is usually solved through chemical modification of the graphene sheets [1,13]: graphite is initially oxidized to graphite oxide, which can be exfoliated in water to produce graphene oxide (GO). The GO chemical or thermal restoration yields reduced graphene oxide (RGO). The problem is that RGO retains some of the defects of the GO and lacks the unique properties of pristine, unfunctionalized graphene.

Oxidative debris are produced during the oxidation process [35]. For all these reasons noncovalently functionalized graphene was recently used [13]. A simple and effective method to disperse pristine graphene using a polyvinylpyrrolidone (PVP) of low molecular weight (10,000 g·mol^−1^) was proposed [36]. Thanks to surface noncovalent functionalization the graphene dispersion were stabilized against aggregation in a range of solvents including water.

In the present study, these pristine graphene dispersions stabilized with low molecular weight PVP (10,000 g·mol^−1^) were suspended in a high molecular weight (1,300,000 g·mol^−1^) PVP solution that was successfully electrospun. The paper shows that graphene addition does not lower the very high sound absorption coefficient α, close to unity, in the frequency range 200–800 Hz, previously reported by the authors [26] for pure PVP mats. The main focus of the paper is, however, on the non-monotonous effect that the addition of graphene has on the structure (fibre diameter and polymer chain conformation), acoustical properties and size of electrospun mat. An explanation of the effect on structure and electrospun mats size is proposed based on the currently accepted mechanism of electrospinning deposition.

## 2. Experimental

### 2.1. Materials

Graphite flakes (CAS Number 7782-42-5), poly(vinyl pyrrolidone) (PVP) (MW:10,000 g·mol^−1^) and Poly(vinyl pyrrolidone) (PVP) (MW:1,300,000 g·mol^−1^) were purchased from Sigma Aldrich (Saint Louis, MO, USA).

### 2.2. Stable Graphene Dispersion

Stable graphene dispersions were prepared by adding 40 mg·mL^−1^ of flakes graphite to the PVP (MW:10,000 g·mol^−1^) solution in alcohol (10 mg·mL^−1^). The solution was tip sonicated using a Q500 sonicator (QSonica) with an output power of 150 W for 1 h at room temperature. The dispersions were then centrifuged at ~7000 rpm for 4 h to remove aggregates. It has been demonstrated in the past that this method yields noncovalently functionalized graphene dispersions [36].

### 2.3. Sample Preparation for the Electrospinning

Alcoholic (ethanol) PVP (MW:1,300,000 g·mol^−1^) solutions (15 wt.%) with different graphene concentrations (from 0.03 to 7.68 mg·mL^−1^) were prepared. In the following the electrospun samples will be distinguished with the acronyms: PVP-G0.03, PVP-G0.12, PVP-G0.48, PVP-G1.92, PVP-G7.68, where the number that follows G is the concentration of graphene in the electrospun solution. The scheme of the electrospinning apparatus is reported in Figure 1a together with the laboratory setup (Figure 1b).

The solutions were loaded into hypodermic syringes fitted with blunt needles. A syringe pump (Harvard Apparatus, Pump 11 Plus) was used to control the flow rate of the solution (0.200 mL min^−1^). The applied voltage was 21 kV. Fibres were collected on a grounded metal collector (wrapped with a copper foil) which was 39 cm away from the needle at room temperature and relative humidity 45 ± 10%. The as-prepared electrospun non-woven mats were dried out at 100 °C for 90 min and stored in a desiccator.

### 2.4. Characterization and Analysis

#### 2.4.1. Raman Microspectroscopy

A confocal Raman microscope Jasco, NRS-3100 was used to obtain Raman spectra by Spectra Manager^TM^ software (v1.54, Jasco Corporation, Tokyo, Japan, 2009). The 514 nm line of an air-cooled Ar^+^ laser (MellesGriot, 35 LAP431 220), was injected into an integrated Olympus microscope and focused to a spot diameter of approximately 3 μm by a 20× objective with a final 4 mW power at the sample. A holographic notch filter was used to reject the excitation laser line. The Raman backscattering was collected using a 0.1 mm slit and a diffraction lattice of 1200 grooves/mm, corresponding to an average spectral resolution of 8 cm^−1^. Both mats and starting solutions have been investigated. Solutions were left evaporating on Si substrates and it took 60 s to collect a complete data set by a Peltier-cooled 1024 × 128 pixel CCD photon detector (Andor DU401BVI). Raman measurements were at least triplicated for scope of reproducibility. Wavelength calibration was performed by using cyclohexane as a standard.

#### 2.4.2. Atomic Force Microscopy

The morphological characteristics of the flakes were analysed by atomic force microscopy (AFM) using a microscope Digital Instruments Nanoscope IIIawith WSxM software (v5.0 develop 9.1, Nanotec Electrónica S.L., Madrid, Spain, 2007), equipped with a sharpened silicon tip having an apical curvature radius of 5 nm. The AFM images were acquired in tapping mode under ambient conditions, with a scan size and rate of 2 μm and 1 Hz, respectively. After performing deconvolution on each AFM image, in order to minimize the tip size effect, the three-dimensional view of the deposits was reconstructed and the thickness was evaluated by means of an image processing technique.

#### 2.4.3. Scanning Electron Microscopy

Scanning Electron Microscopy (SEM), with FEI Inspect and by using Image-Pro Plus (v7.0, Media Cybernetics, Rockville, MD, USA, 2009), observations, so as reported elsewhere [26]. The statistical analysis was performed by measuring the diameter of hundreds of fibres of ten different micrographs, relative to at least three different samples (for each graphene content).

#### 2.4.4. Sound Absorption Coefficient

The sound absorption coefficients were measured through an impedance tube so as reported in the previous paper [26] according to ASTM E1050 and ISO 10534-2, through the impedance tube and Pulse LabShop (v6.1.5.65, Brüel&Kjær, Nærum, Denmark, 2002) shown in Figure 2. According to Chung and Blaser’s [37] results, the complex sound reflection coefficient, calculated from the corrected acoustic transfer function *H*_12_ is:
(1)$$R=R_{r}+jR_{i}=\frac{H_{12}-e^{-jks}}{e^{jks}-H_{12}}e^{2jk\left(s+L\right)}$$
where *R_r_* and *R_i_* are respectively real and imaginary part of complex acoustic reflection coefficient (*R*), *k* is wave number and it is equal to 2*πf*/*c* (*f* is the working frequency, *c* is the sound speed in the air), *L* distance from the test sample to the centre of the nearest microphone, *s* centre-to-centre spacing between microphones and *j* = √(−1). From Equation (1) it is possible to calculate the sound absorption coefficient at normal incidence as a function of frequency [37,38] and the normal acoustic specific impedance respectively as:
(2)$$\alpha =1-{\left|R\right|}^{2}=1-R_{r}^{2}-R_{i}^{2}$$
(3)$$\frac{Z}{Z_{0}}=\frac{r}{\rho_{0}c_{0}}+\frac{jx}{\rho_{0}c_{0}}=\left(\frac{1+R}{1-R}\right)$$
where $Z_{0}=\rho_{0}c_{0}$ is the characteristic impedance of the medium with *p*_0_ and *c*_0_ respectively density and speed of sound in the air, $r/\rho_{0}c_{0}$ is the normal specific acoustic resistance ratio and $jx/\rho_{0}c_{0}$ is the normal specific acoustic reactance ratio.

## 3. Results and Discussion

### 3.1. Structure of Graphene

Graphite was exfoliated in the presence of low molecular weight PVP (MW: 10,000 g·mol^−1^) so as described in the experimental section with a process recently proposed in the literature [36]. The structure of the obtained particles was investigated by means of Atomic Force Microscopy (AFM) (Santa Clara, CA, USA) and Raman microspectroscopy.

Figure 3a shows a representative AFM image of an isolated flake. The particle height profile reported in Figure 3b shows that the average thickness of this flake is less than 1 nm. An average thickness value for the particles in the sample was obtained by analysing the profiles of 12 isolated flakes, finding the value of (1.1 ± 0.3) nm, where the error is the standard deviation. This indicates that multilayer graphene flakes were obtained.

The Raman spectra (on excitation with wavelength of 514.5 nm) of graphite, exfoliated graphite (labelled as “NCFG” in the Figures) and pure PVP are reported in Figure 4a,b. In the low frequency region of the graphite and exfoliated graphite spectra, the characteristic graphene G band at 1587 cm^−1^ and a G′ band around 2700 cm^−1^ are present. The ratio between the G and G′ band (2:1) and the shape of the G′ band in the “NCFG” spectrum indicates a three-layer structure of graphene [38,39]. The NCFG spectrum shows also all the characteristic bands of PVP. Therefore the flakes are functionalised with PVP of low molecular weight so as expected [36].

### 3.2. Structure of Electrospun Mats

The electrospun mats were produced in thin layers (about 1 g weight) corresponding to an electrospun solution volume of 8 mL.

Figure 5 shows how the area of the deposit on the target changes in the course of electrospinning. In the case of PVP the first layers deposited on the target do have a small diameter that, as usually occurs, increases during electrospinning. However, in this regard, Figure 5 shows that the graphene addition strongly affects the deposition process. At high graphene content (PVP-G7.68) the curve resembles the PVP one. On the contrary, small additions of graphene (PVP-G0.03, PVP-G0.12) make the deposit area to be large as soon as the electrospinning starts. The curves for PVP and PVP-G7.68 show an area increase over 12′ time that is 30% of the time (40′) necessary to evacuate the syringe and to make a single PVP sheet. This means that the first additions of graphene flakes (samples PVP-G0.03, PVP-G0.12) allow to produce sheets of much more uniform thickness. The non-uniformity progressively increases again at higher graphene content. A non-monotonous trend is observed, therefore, at increasing graphene flakes content.

Figure 6a,b show the SEM micrograph of sample PVP-G1.92 which is representative of all the samples. A particular of it, at higher magnification, is also shown in Figure 6b. As can be seen grooves are present at the surface whose dimension corresponds to the size of graphene flakes shown in Figure 3.

Figure 7 shows how the fibre diameter, measured from the SEM micrographs, changes as a function of the graphene content. A minimum is observed at the lower graphene content (PVP-G0.03, PVP-G0.12). A non-monotonous trend with graphene flakes content is, therefore, observed also for the fibre diameter.

Figure 8 shows the Raman spectra of the PVP mats. Because of the very low graphene contents, a very weak G band (1586 cm^−1^) is detectable only for the highest-graphene content mat, (Figure 4). Moreover the non-monotone trend observed in the results reported in Figure 5 and Figure 7 has an interesting counterpart with Raman features related to PVP conformation. Indeed, upon graphene addition, there is a non-monotone trend (decreasing till sample PVP-G0.48 and then increasing till sample PVP-G7.68 of both the Raman bands around at 1312 and 1030 cm^−1^ (Figure 8). Particularly, the 1030 cm^−1^ frequency is due to both a C-C stretching and to CH_2_ rocking [40], that are dependent on polymer main-chain conformation [41]. Therefore, it can be suggested that graphene addition up to sample PVP-G0.48 produces a conformational change that returns to the original conformation in samples PVP-G1.92 and PVP-G7.68. This structural modification might be tentatively related to the non-monotone trend observed in Figure 5 and Figure 7. Differently from main-chain markers, lactam C=O stretching (related to pyrrolidone side chain) is only slightly affected by graphene addition (2 cm^−1^ in the presence of graphene), much lower than that observed in silver suspension in PVP, where Ag^+^-induced redshifts are observed ranging from 16 to 27 cm^−1^ for different PVP molecular weights [42].

In order to explain the results it is worth reminding that, in the electrospinning apparatus, fibres are formed under the action of an electrostatic force. Basically, an electrospinning system, whose scheme is reported in Figure 1, consists [17,18] of three major components: a high voltage power supply, a spinneret (e.g., a pipette tip) and a grounded collecting plate (usually a metal screen, plate, or rotating mandrel). In the electrospinning process, a polymer solution held by its surface tension at the end of a capillary tube is subjected to a strong electric field and, as a consequence, an electric charge is induced on the liquid surface [17,18,43,44]. This causes deformation of the liquid droplet into the so called “Taylor cone” [45,46]. When the surface tension is overcome by the electric field, a charged jet is ejected from the tip of the Taylor cone. The jet is stable only near the tip of the spinneret. Very soon it becomes unstable so as depicted in Figure 1 acquiring a more or less complex whipping movement, during which electrical forces stretch and thin it by very large ratios. During the flight the solvent evaporates and polymer fibres are, finally, deposited on the target. Of course the level of instability during the jet flight determines the mat structure.

Recently the origin of the instability was explained [47]. Near the tip of the syringe the longitudinal stress caused by the external electric field, acting on the charge carried by the jet, stabilizes for some distance the jet, that runs straight and is considered to have a viscoelastic behaviour. However as soon as a fluid element departs accidentally from the trajectory, a lateral perturbation grows in response to the repulsive forces between adjacent elements of charge carried by the jet. The motion of segments of the jet, then, grows rapidly into an electrically driven bending instability.

According to the above description, a possible role of graphene can be proposed so as sketched in Figure 9. Indeed, the instability is caused by the surface charge of the fluid elements that constitute the jet. The presence of graphene may give rise to a greater surface charge and, therefore, larger and more complex whipping movements. This may well explain the increase of the deposit area (Figure 5) at short times observed in the case of the samples with low graphene content. Also the fibre diameter reduction (Figure 7) may be the consequence of the greater stretching. However if the graphene content is increased too much other effects become dominant. It is worth remembering that the success in the exfoliation of graphite in the presence of PVP, in fact, is due to good interactions of the graphene sheets with PVP [36]. Excellent load transfer was demonstrated [36] for PVP stabilized graphene/PVP composite after polymerization. The improvement of the viscoelastic properties may well cause, at higher graphene content, a lower jet instability, lower stretching and, as a consequence, larger fibre diameters. The conformational changes suggested by Raman spectra in correspondence of the lower graphene contents may so, also, be explained.

### 3.3. Sound Absorption Properties of Electrospun Mats

As it is known sound waves are longitudinal mechanical waves propagating at speeds and frequencies depending on the acoustical properties of the medium. When a sound wave impacts upon the surface of a solid body, some portion of its energy is reflected, some absorbed and the rest transmitted through the body. An absorption coefficient, α_a_, is defined simply as the ratio of sound energy absorbed to the incident one. It changes with the frequency of the sound wave. The acoustic properties of the samples were measured through the acoustic impedance tube.

In Figure 10 the plots of the sound absorption coefficient versus the frequency are reported for the electrospun mats of PVP and PVP added with graphene (PVP-G0.12, PVP-G0.48, PVP-G7.68). All the mats were obtained by stacking on each other the sheets obtained through electrospinning till a compressive mass of about 7.5 ± 0.5 g.

Bell shaped curves, with very high absorption coefficient at the maximum, were, in all cases, obtained so as reported by the authors for pure PVP in a previous paper [26]. Figure 10 shows that the bell shaped curves become sharper and the maximum shifts to higher frequency at low graphene addition. The opposite is observed when the graphene content is further increased. Therefore the shape (broader or sharper) and frequency of maximum depends in a non-monotonous manner on the graphene content so as observed for the results reported in Figure 5, Figure 7 and Figure 8. This offers a practical advantage to allow to tune the acoustical response of the mats at constant total mass.

## 4. Conclusions

Tip sonication of graphite alcoholic suspensions in the presence of low molecular weight PVP is confirmed effective to produce graphene, with a three layer structure functionalised with low molecular weight PVP.

The graphene addition affects in a non-monotonous manner the electrospinning process. Bending instability of electrospun jet appears to increase at low graphene concentration with the result of greater fibre stretching that leads to lower fibre diameter and possibly conformational changes of PVP. Further increase of graphene content seems having the opposite effect leading to greater fibre diameter and Raman spectra similar to the pure PVP electrospun mats. All this appears to be related to the acoustic properties that also vary in a non-monotonous way. It is worth underlining that, at knowledge of the authors, this is the first time such non-monotonous effects on the addition of graphene are described.

## Figures and Tables

**Figure 1 materials-12-00108-f001:**
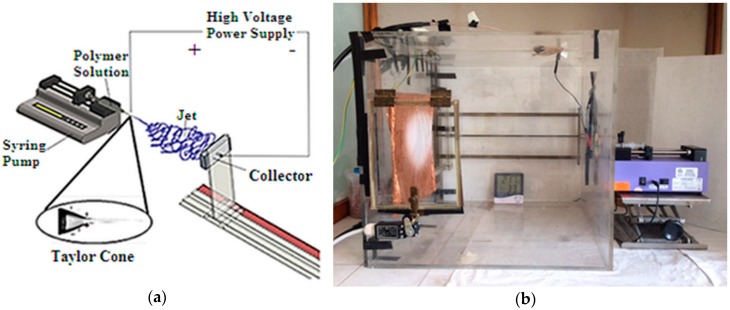
Electrospinning apparatus. (**a**) Schematic setup; (**b**) Laboratory setup.

**Figure 2 materials-12-00108-f002:**
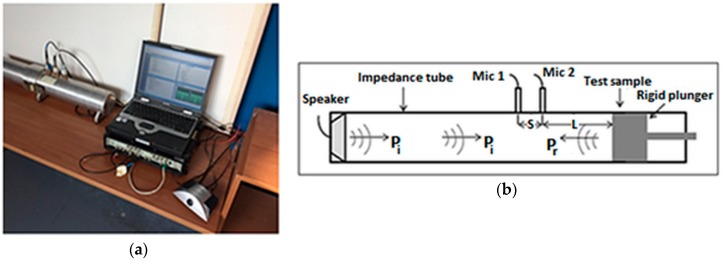
Acoustic sound absorption coefficient measurement. (**a**) Measurement setup in the laboratory; (**b**) Schematic of two-microphone impedance tube method.

**Figure 3 materials-12-00108-f003:**
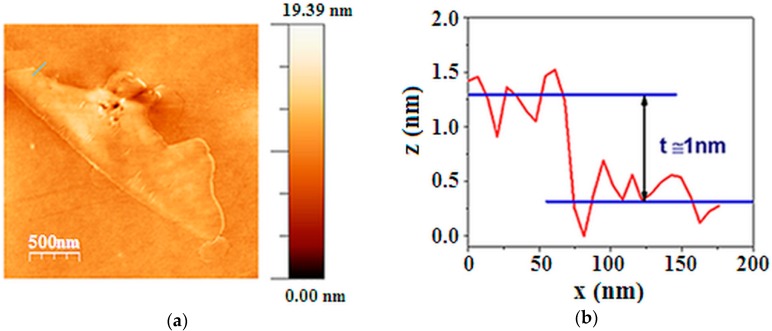
(**a**) 2D AFM image of graphenes (AFM Tapping mode Tip Radius < 10 nm); (**b**) Graphenes height profile.

**Figure 4 materials-12-00108-f004:**
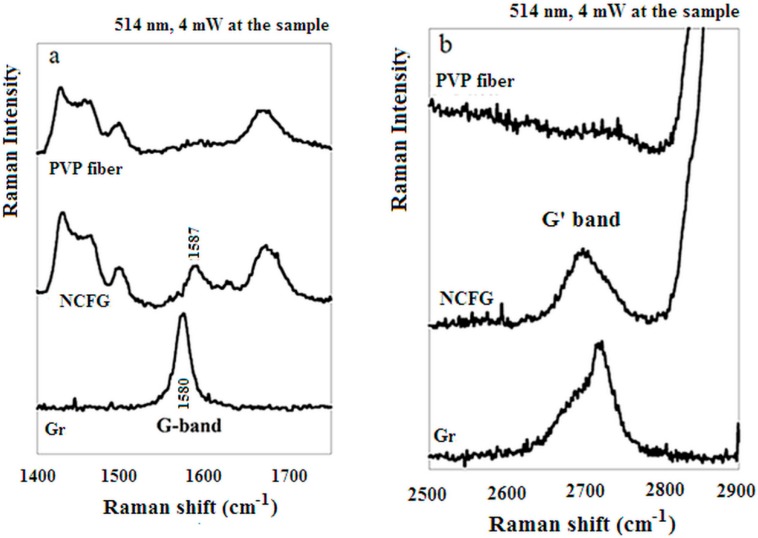
Raman spectra of PVP fibre, Noncovalently Functionalized Graphene (NCFG) and Graphite (Gr). (**a**) Raman shift range from 1400 to 1700 cm^−1^; (**b**) Raman shift range from 2500 to 2900 cm^−1^.

**Figure 5 materials-12-00108-f005:**
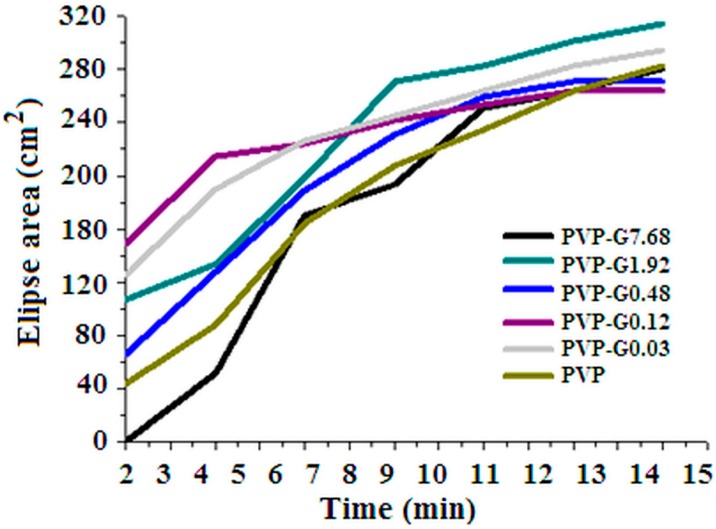
Deposit area as a function of time for samples PVP-G0.03, PVP-G0.12, PVP-G0.48, PVP-G1.92, PVP-G7.68.

**Figure 6 materials-12-00108-f006:**
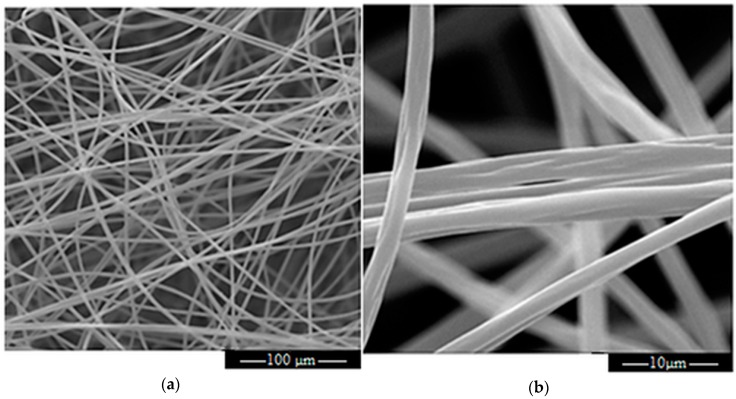
(**a**) SEM micrograph of PVP-G1.92; (**b**) SEM micrograph of PVP-G1.92 at an higher magnification.

**Figure 7 materials-12-00108-f007:**
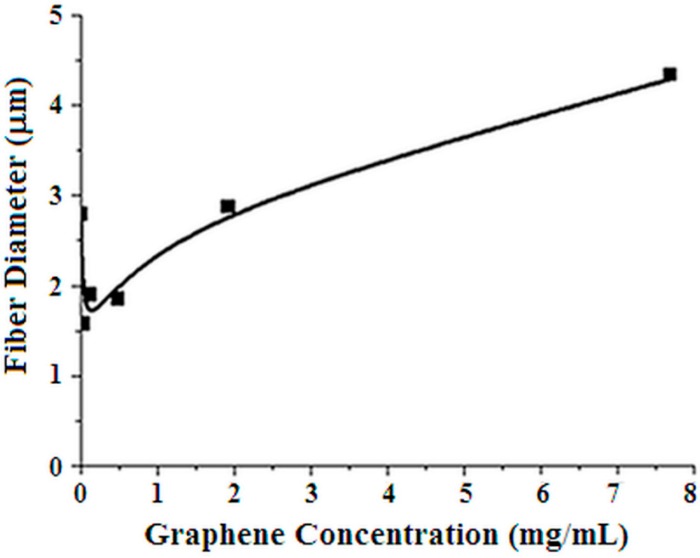
Plot of fibre diameter as a function of graphene content in the electrospun solution.

**Figure 8 materials-12-00108-f008:**
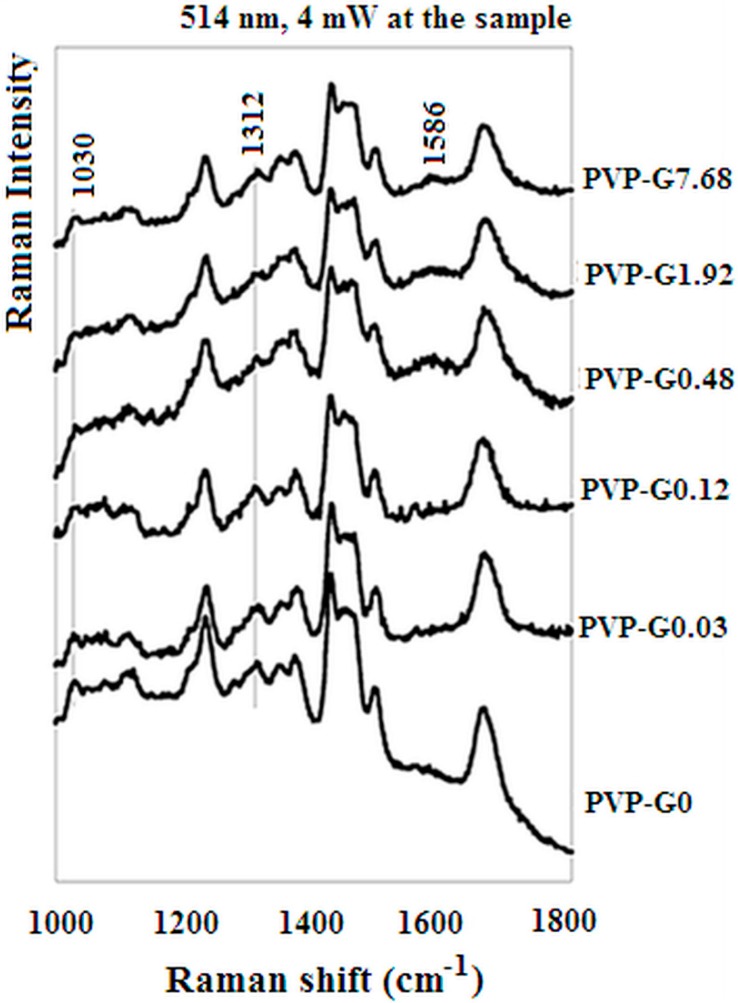
Raman spectra of the PVP tissues with different graphene content (PVP-G0.03, PVP-G0.12, PVP-G0.48, PVP-G1.92, PVP-G7.68) (excitation wavelength 514 nm, 4 mW at the sample).

**Figure 9 materials-12-00108-f009:**
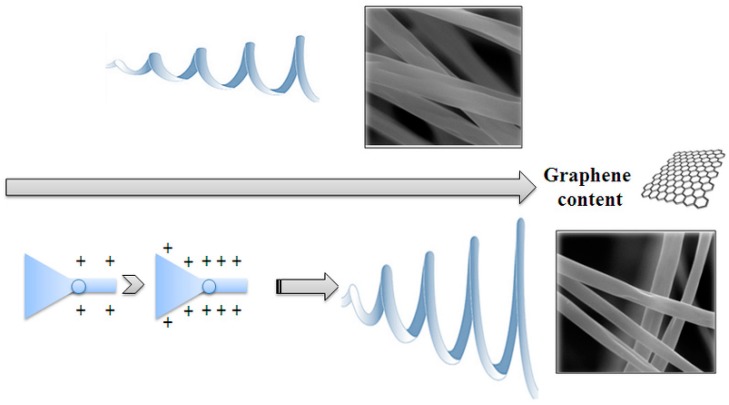
Sketch of the effect of noncovalently functionalised graphene addition on the structure of the electrospun mat.

**Figure 10 materials-12-00108-f010:**
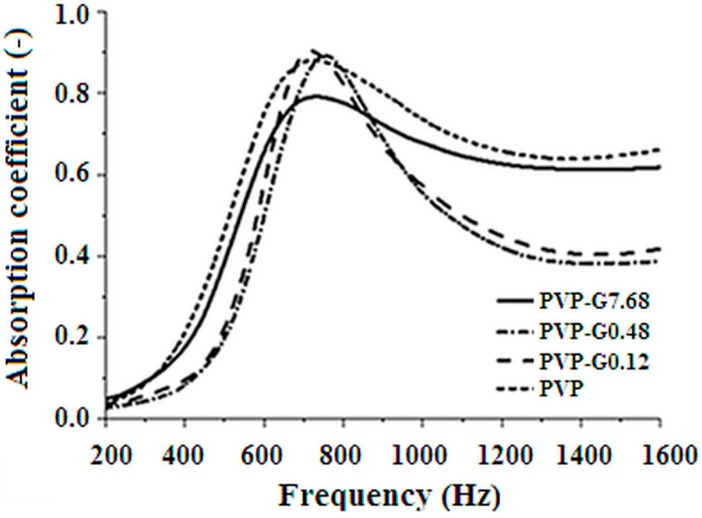
Sound absorption coefficient as a function of frequency.

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
