# Peer review of "Non Monotonous Effects of Noncovalently Functionalized Graphene Addition on the Structure and Sound Absorption Properties of Polyvinylpyrrolidone (1300 kDa) Electrospun Mats"

_materials, 2018, doi:10.3390/ma12010108_

Reviewer 1 Report

Please highlight the advances and advantages with regard to previous works related to the electrospinning fabrication process, especially in the conclusions.
Please provide an actual image of the electrospinning setup to complement Figure 1.
The authors have stated that the average thickness of the flakes is less than 1nm. What the authors mean by "average"? how many flakes were measured?
Text in Figures 4, 5, 7 and 10 is a little bit blurry. Please improve them.

There are some typos along the text requiring for a careful editing. Just to mention some of them:
Double space in line 65 after known.
Missing space in 117.
Missing full stop in 119.
Double space in 214.
Missing space in 262.
Double space in 280.

Author Response

Response to Reviewer 1 Comments

The authors are grateful to the reviewers for the valuable comments that allowed to improve the quality of the paper. In the following a description of how the authors took advantage of the reviewers comments.

The authors hope to have fully satisfied all points raised by the reviewers.

Point 1: Please highlight the advances and advantages with regard to previous works related to the electrospinning fabrication process, especially in the conclusions.

Response 1: The originality of the paper was underlined in the conclusion section (lines 374-376).

Point 2: Please provide an actual image of the electrospinning setup to complement Figure 1.

Response 2: The setup was added as Fig. 1b.

Point 3: The authors have stated that the average thickness of the flakes is less than 1nm. What the authors mean by "average"? how many flakes were measured?

Response 1: The statement was clarified (lines 237-241).

Point 4: Text in Figures 4, 5, 7 and 10 is a little bit blurry. Please improve them.

Response 4: All the figures were improved.

Point 5: There are some typos along the text requiring for a careful editing. Just to mention some of them:
Double space in line 65 after known.
Missing space in 117.
Missing full stop in 119.
Double space in 214.
Missing space in 262.
Double space in 280.

Response 5: The text was carefully checked and all typos were eliminated.

Reviewer 2 Report

This paper investigates the non-monotonous effects of noncovalently functionalized graphene addition on the structural and sound absorption properties of PVP. The paper demonstrates merits compared to previous studies. It could be considered for publication if minor revisions can be made to address the following comments: 1)In addition to the observations from the figures, the author may need to give necessary analysis on the mechanisms that underpin the phenomena; 2) The quality of the figures may need to be improved. Some of them are not clear enough; 3) For Eqs.  (1) and (2), the authors may need to give details for every symbol used. This may help readers to understand them.

Author Response

Response to Reviewer 2 Comments

The authors are grateful to the reviewers for the valuable comments that allowed to improve the quality of the paper. In the following a description of how the authors took advantage of the reviewers comments.

The authors hope to have fully satisfied all points raised by the reviewers.

This paper investigates the non-monotonous effects of noncovalently functionalized graphene addition on the structural and sound absorption properties of PVP. The paper demonstrates merits compared to previous studies. It could be considered for publication if minor revisions can be made to address the following comments.

Point 1: In addition to the observations from the figures, the author may need to give necessary analysis on the mechanisms that underpin the phenomena.

Response 1: An explanation had been given by the authors for the non-monotonous effect of graphene addition on the electrospinning deposition and mats structure (fiber diameter and Raman spectra) on lines 307-342. Owing to lack of competence in acoustical curves modelling, the acoustic impedance results were not better correlated to the above reminded explanation. However the authors point out that the aim of the paper was to show the existence of a non monotonous effect that may reflect also on the properties, like the acoustical ones. The authors hope to have satisfied the reviewer good point.

Point 2: The quality of the figures may need to be improved. Some of them are not clear enough;

Response 2: All the figures were improved.

Point 3: For Eqs. (1) and (2), the authors may need to give details for every symbol used. This may help readers to understand them.

Response 3: The text was improved (lines 197-217).

Reviewer 3 Report

In this work by Giuseppe Rosario Del Sorbo and co-workers the authors have presented how they used graphene-reinforced PVP mats for sound absorption. Development of sound absorption materials is interesting and important because of the growing problem of sound pollution all around the world. Please find my suggestions below, which could improve the shape of this work:

1) I suggest proof-reading of the work by a native speaker. There are some typos and errors present, which make it difficult to understand the concept. Examples of errors: "seam" (line 27), "it's" (line 40), "composited" (line 49), "flakes graphite" (line 107), "grafite" (many instances), etc. I would also suggest make the statements a bit less intense. Example: "raised very great interest", "enormous attention"

2) According to your introduction, the only reason for incorporation of graphene into PVP here is its ability to retard flames. Could you please disclose other reasons for adding graphene into the PVP for this purpose? It would enable readers to get a clearer view about the potential impact of this work.

3) Whether the outcome of sonication is positive or negative is very much dependent on the power of the sonicator. I suggest including this value. 

4) Could you please describe why you did not suspend graphene with 1.3M PVP?

5) Have you tried to analyze the graphene by TEM instead of AFM? 

6) Caption to the Fig. 4 does not correspond to what is shown. You have written "Raman spectra of grafite", but there are other samples shown in the plot. This section and description of Figure 8 should be improved because they lack clarity.

7) How many samples have been produced for each graphene loading? I strongly suggest repeating the study with more than one sample for every parameter combination because the local minimum in Fig. 7 can be a statistical error.

8) Fig. 9 could also be improved. You may try to depict how stable and unstable jet looks like. Simultaneously, I also suggest speaking about this in terms of "stability" nor "instability" because in your case it creates double negatives which are harder to follow. For instance: "increasing jet instability" (Fig. 9)

Author Response

Response to Reviewer 3 Comments

The authors are grateful to the reviewers for the valuable comments that allowed to improve the quality of the paper. In the following a description of how the authors took advantage of the reviewers comments.

The authors hope to have fully satisfied all points raised by the reviewers.

In this work by Giuseppe Rosario Del Sorbo and co-workers the authors have presented how they used graphene-reinforced PVP mats for sound absorption. Development of sound absorption materials is interesting and important because of the growing problem of sound pollution all around the world. Please find my suggestions below, which could improve the shape of this work.

Point 1: I suggest proof-reading of the work by a native speaker. There are some typos and errors present, which make it difficult to understand the concept. Examples of errors: "seam" (line 27), "it's" (line 40), "composited" (line 49), "flakes graphite" (line 107), "grafite" (many instances), etc. I would also suggest make the statements a bit less intense. Example: "raised very great interest", "enormous attention".

Response 1: The English was checked and all typos and errors were corrected.

Point 2: According to your introduction, the only reason for incorporation of graphene into PVP here is its ability to retard flames. Could you please disclose other reasons for adding graphene into the PVP for this purpose? It would enable readers to get a clearer view about the potential impact of this work;

Response 2: This is a good point. The main reason was of course to obtain a better behavior in case of fire. However also improving mechanical strength would be of interest for some applications of these soundproofing materials. This interest was added on lines 84-85.

Point 3: Whether the outcome of sonication is positive or negative is very much dependent on the power of the sonicator. I suggest including this value;

Response 3: The sonicator output power was 150 W. This information was added to the paper (line 122).

Point 4: Could you please describe why you did not suspend graphene with 1.3M PVP?

Response 4: We simply followed the procedure proposed by Wajid, who demonstrated that exfoliation occurred successfully in the presence of lower molar mass PVP. Of course the authors agree that trying to exfoliate graphite in the presence of the greater molar mass PVP (the same that is electrospun) could be of interest.

Point 5: Have you tried to analyze the graphene by TEM instead of AFM? 

Response 5: Unfortunately we had no availability of TEM.

Point 6: Caption to the Fig. 4 does not correspond to what is shown. You have written "Raman spectra of grafite", but there are other samples shown in the plot. This section and description of Figure 8 should be improved because they lack clarity;

Response 6: The Figure was improved and the caption corrected.

Point 7: How many samples have been produced for each graphene loading? I strongly suggest repeating the study with more than one sample for every parameter combination because the local minimum in Fig. 7 can be a statistical error;

Response 7: The electrospun mats were produced (see line 258-259) in thin layers (about 1 g weight). Therefore, in order to prepare the samples for impedence tube analysis at least 7-8 such thin layers had to be prepared. So as reported  in section 2.4.3 the morphology and fibers average diameter were characterized by off line image analysis techniques of the acquired images (Image-Pro Plus). The statistical analysis was performed by measuring the diameter of hundreds of fibers of ten different micrographs. The ten micrographs were relative to at least three different samples (for each graphene content). This specification was added on lines 175-176.

Point 8: Fig. 9 could also be improved. You may try to depict how stable and unstable jet looks like. Simultaneously, I also suggest speaking about this in terms of "stability" nor "instability" because in your case it creates double negatives which are harder to follow. For instance: "increasing jet instability" (Fig. 9);

Response 8: We followed the revewer very good suggestion by very much improving Fig.9. By this way there was no need to speak of “instability” or “stability”. Accordingly also the graphical abstract was changed.

Round  2

Reviewer 3 Report

Thank you for including my suggestions. I can now recommend publication of these results.